# Learning to Rank with Top-$K$ Fairness

**Boyang Zhang**                                                          *bzhan29@lsu.edu*
*Computer Science & Engineering, Louisiana State University*

**Quanqi Hu**                                                          *quanqi-hu@tamu.edu*
*Computer Science & Engineering, Texas A&M University*

**Mingxuan Sun**[*]                                                          *msun11@lsu.edu*
*Computer Science & Engineering, Louisiana State University*

**Qihang Lin**                                                          *qihang-lin@uiowa.edu*
*Business Analytics, University of Iowa*

**Tianbao Yang**                                                          *tianbao-yang@tamu.edu*
*Computer Science & Engineering, Texas A&M University*

**Reviewed on OpenReview:** *https://openreview.net/forum?id=SSPCc39XvO*

## Abstract

Fairness in ranking models is crucial, as disparities in exposure can disproportionately affect protected groups. Most fairness-aware ranking systems focus on ensuring comparable average exposure for groups across the entire ranked list, which may not fully address real-world concerns. For example, when a ranking model is used for allocating resources among candidates or disaster hotspots, decision-makers often prioritize only the top-$K$ ranked items, while the ranking beyond top-$K$ becomes less relevant. In this paper, we propose a list-wise learning-to-rank framework that addresses the issues of inequalities in top-$K$ rankings at training time. Specifically, we propose a top-$K$ exposure disparity measure that extends the classic exposure disparity metric in a ranked list. We then learn a ranker to balance relevance and fairness in top-$K$ rankings. Since direct top-$K$ selection is computationally expensive for a large number of items, we transform the non-differentiable selection process into a differentiable objective function and develop efficient stochastic optimization algorithms to achieve both high accuracy and sufficient fairness. Extensive experiments demonstrate that our method outperforms existing methods.

## 1 Introduction

Top-$K$ fairness in ranking has become a critical concern since unfair rankings lead to inequalities such as unequal business opportunities, educational placements, and resource allocation (Kulshrestha et al., 2017; Mohler et al., 2018; Shang et al., 2020). Ranking models used for decision-making typically provide more exposure to top-ranked items than those ranked lower (Singh & Joachims, 2018). For example, in educational systems, funding agencies allocate more resources to top-ranked schools (Darling-Hammond, 2001). In hiring systems, only the top-$K$ candidates may be interviewed (Kweon et al., 2024). Similarly, in disaster response or predictive policing (Mohler et al., 2018), top-$K$ hotspot rankings determine where limited resources are deployed. In such settings, fairness in the top-$K$ results has direct and significant real-world impact. Due to factors like historical discrimination, items in the protected group that possess a specific attribute, such as gender, are often under-represented within the training dataset. This can lead the model to generate rankings that exhibit substantial disparities in exposure between groups.

---

[*]Corresponding author

Fairness in ranking differs significantly from traditional fairness metrics in classification, as it requires considering position bias. Traditional fairness literature mainly focuses on ensuring equal classification outcomes, such as equalized odds and demographic parity (Hardt et al., 2016; Zafar et al., 2017). Pairwise fairness metrics (Abdollahpouri et al., 2017; Beutel et al., 2019; Narasimhan et al., 2020; Fabris et al., 2023), which are derived from fairness definitions in binary classification, focus on preserving the pairwise relative order of items according to their relevance scores, irrespective of group membership. However, these metrics often overlook the position bias inherent in ranking, where top-ranked items receive more attention. In contrast, list-wise metrics such as Biega et al. (2018); Singh & Joachims (2018); Zehlike & Castillo (2020) better address fairness in ranking. While some (Biega et al., 2018) focus on individual fairness, the majority of them emphasize group fairness, ensuring that different groups receive similar average exposure.

While post-processing methods (Zehlike et al., 2017; Biega et al., 2018; Asudeh et al., 2019; Mehrotra & Vishnoi, 2022) are proposed to address fairness by re-ordering ranked lists, these approaches face two key limitations: (1) suboptimal trade-offs between relevance and fairness, and (2) reliance on group labels during testing. First, post-processing methods adjust rankings after the model has been trained, meaning they do not incorporate fairness constraints within the model itself, which makes it difficult to achieve the best ranking quality for a given fairness level or to achieve the highest fairness for a fixed ranking quality. Second, these methods access to sensitive group labels during testing, which can limit generalization and raise privacy concerns, especially when such labels are unavailable or sensitive.

In contrast, in-processing methods such as Zehlike & Castillo (2020) that integrate fairness directly into the learning-to-rank process during training ensure Pareto efficiency. Specifically, in-processing methods can be viewed as solving a constrained optimization model where the quality of ranking is optimized subject to a constraint that bounds the level of unfairness. This means the in-process methods guarantee that no further improvement in fairness can be made without sacrificing ranking quality. While existing in-processing methods (Zhu et al., 2021; Memarrast et al., 2023) account for position bias, they do not emphasize fairness in critical top-$K$ positions. This limitation underscores the need for in-processing approaches that explicitly ensure fairness within the top-$K$ rankings.

In this paper, we propose a list-wise learning-to-rank framework that addresses the issue of inequalities in top-$K$ rankings at training time for the first time. Specifically, we introduce a novel **top-$K$ exposure disparity** metric, which is an extension of the group exposure disparity in a ranked list (Singh & Joachims, 2018; Zehlike & Castillo, 2020). We then learn a ranker that optimizes the list to balance ranking relevance and exposure disparities at top-$K$ positions. A direct top-$K$ selection process, such as a naive approach based on sorting the whole list, is computationally expensive for a large number of items. To this end, we transform the non-differentiable top-$K$ selection into a differentiable objective function and develop efficient stochastic algorithms.

To empirically validate the effectiveness of our method, we conduct a comprehensive set of experiments using popular benchmark datasets. The experimental results demonstrate that our method not only achieves high ranking accuracy but also significantly alleviates exposure disparities at top-$K$ positions when compared to several state-of-the-art methods. To the best of our knowledge, this is the first time an in-processing learning-to-rank framework is proposed to address both relevance and fairness in top-$K$ rankings with a provable convergence guarantee.

## 2 Related Work

Ranking fairness metrics based on pairwise comparisons (Abdollahpouri et al., 2017; Beutel et al., 2019; Narasimhan et al., 2020; Fabris et al., 2023) are proposed to ensure the relative order of a pair is consistent with certain fairness principles. For example, a ranking algorithm is considered fair if the likelihood of a clicked item being ranked higher than another relevant unclicked item is equal across groups, provided both items have received the same level of engagement (Beutel et al., 2019). In contrast, list-wise approaches optimize fairness across the entire ranking list by ensuring balanced exposure and relevance for all items (Singh & Joachims, 2018; Yang & Stoyanovich, 2017; Zehlike & Castillo, 2020; Kotary et al., 2022). For example, a statistical parity-based measure Yang & Stoyanovich (2017) is introduced to calculate the difference in the distribution of various groups across different prefixes of the ranking. Studies such as Singh & Joachims

(2018); Zehlike & Castillo (2020) define equal exposure fairness, aiming to equalize the average exposures between minority and majority groups. Some other methods (Chakraborty et al., 2022; Zhao et al., 2023) address fairness in ranking aggregation, a different area from learning-to-rank.

Ranking fairness can be addressed through pre-, in-, and post-processing approaches. Pre-processing methods aim to prevent biased models, but creating an unbiased training set is complex and can result in reverse discrimination (Zehlike & Castillo, 2020). Post-processing methods (Islam et al., 2023; Gao et al., 2022; Yang et al., 2023; Mehrotra & Vishnoi, 2022; Zehlike et al., 2022; 2017; Biega et al., 2018; Asudeh et al., 2019; Vardasbi et al., 2024; Gorantla et al., 2024) are developed to re-rank a list to satisfy specific fairness criteria. Specifically, The FA*IR algorithm, introduced in Zehlike et al. (2017), adjusts ranking lists to ensure that the ratio of items from the protected group in each prefix of the top-$K$ rankings meets or exceeds a specified minimum threshold. Additionally, the algorithm is enhanced to accommodate multiple protected groups (Zehlike et al., 2022). However, the heuristic adjustments are constrained by other models and are only compatible with their specific fairness metric. Moreover, there is no widely accepted definition of top-$K$ fairness nor theoretical guarantee for satisfying fairness constraints (Lahoti et al., 2019).

Post-processing methods such as Celis et al. (2017); Singh & Joachims (2018; 2019); Kotary et al. (2022) aim to determine a probabilistic ranking that maximizes utility while satisfying fairness constraints. For example, a doubly stochastic matrix represents the probability of item $i$ being ranked at position $j$, and the optimal matrix is learned to maximize expected utility subject to group exposure fairness constraints. The matrix is solvable via linear programming, and the sampled rankings achieve exposure fairness in expectation (Singh & Joachims, 2019). These methods are categorized as post-processing because they depend on an underlying predictive model to estimate item relevance. Additionally, these methods aim to estimate the probabilities of each item being ranked at any position, which are computationally expensive.

In-processing methods aim to address fairness directly within the ranking model during training. DELTR (Zehlike & Castillo, 2020) extends ListNet (Cao et al., 2007) to a framework that optimizes ranking accuracy and reduces unfairness, defined as discrepancies in ranking exposure between two groups. Zhu et al. (2020) introduce a debiased personalized ranking model addressing item under-recommendation bias by improving ranking-based statistical parity and equal opportunity. Robust (Memarrast et al., 2023) constructs a minimax game for fair ranking by balancing fairness constraints with utility using distributional robustness principles, achieving fairness-utility trade-offs. MCFR (Wang et al., 2024) introduces a meta-learning framework combining pre- and in-processing techniques with curriculum learning to address fairness across entire ranked lists. However, these approaches concentrate on the entire ranked list rather than prioritizing top-$K$ positions.

In-processing approaches for top-$K$ fairness are lacking. Unlike post-processing methods, we focus on in-processing approaches that provide a theoretical guarantee for satisfying fairness constraints at top-$K$ positions. Different from K-SONG (Qiu et al., 2022), which focuses solely on top-K ranking relevance (NDCG), we propose a stochastic algorithm to optimize a top-$K$ ranking that achieves both high relevance, as measured by NDCG, and sufficient fairness, quantified by reducing exposure disparities between minority and majority groups. Moreover, unlike other in-processing fair ranking frameworks (Memarrast et al., 2023; Zhu et al., 2020), which are not scalable due to their model complexities, our framework is efficiently optimizable and scalable to large datasets.

## 3 Preliminaries

Let $\mathcal{Q}$ denote a set of $N$ queries and $q \in \mathcal{Q}$ denote a query (e.g., a query for document retrieval or a user for recommendation). Let $S_q = \{\mathbf{x}_i^q | i = 1, \ldots, N_q\}$ denote a set of $N_q$ items (e.g., documents, products) to be ranked for $q$, where $\mathbf{x}_i^q$ denote the embedding for each item with respect to query $q$. Let $\mathcal{S}$ be the set of all query-item pairs, i.e., $\mathcal{S} = \{(q, \mathbf{x}_i^q) | q \in \mathcal{Q}, \mathbf{x}_i^q \in \mathcal{S}_q\}$. Let $y_i^q$ denote the relevance score (e.g., a rating) between query $q$ and item $\mathbf{x}_i^q$ and $\mathcal{Y}_q = \{y_i^q\}_{i=1}^{N_q}$ denote the set of all relevance scores for query $q$. For simplicity, we assume that the items in $S_q$ belong to two disjoint groups. Let $\mathcal{S}_a^q \subset S_q$ denote the set of items in the minority group and $\mathcal{S}_b^q \subset S_q$ denote the set of items in the majority group.

Let $h_q(\mathbf{x}; \mathbf{w})$ denote a predictive function that outputs a score for $\mathbf{x}$ with respect to the query $q$ with a higher score leading to a higher rank of $\mathbf{x}$ in an output list. The parameters of the scoring function are denoted

by $\mathbf{w}$ (e.g., a deep neural network). The classical approach to obtaining a good $h_q(\mathbf{x}; \mathbf{w})$ is to optimize $\mathbf{w}$ by maximizing a quality measure on the output lists across all queries or, equivalently, minimizing a loss function that decreases with the output quality.

There exist multiple ways to measure the quality of ranking items in $\mathcal{S}_q$ with respect to a query $q$. One commonly used example is the NDCG measure, which is defined as:

$$\text{NDCG} : \frac{1}{Z_q} \sum_{\mathbf{x}_i^q \in \mathcal{S}_q} \frac{2^{y_i^q} - 1}{\log(1 + r(\mathbf{w}; \mathbf{x}_i^q, \mathcal{S}_q))}. \tag{1}$$

Here, $Z_q$ is a normalization constant and $r(\mathbf{w}; \mathbf{x}, \mathcal{S}_q) \in \{1, 2, \ldots, N_q\}$ is the rank of $\mathbf{x}$ in the set $S_q$ based on $h_q(\mathbf{x}; \mathbf{w})$, that is,

$$r(\mathbf{w}; \mathbf{x}, \mathcal{S}_q) = \sum_{\mathbf{x}' \in \mathcal{S}_q} \mathbb{I}(h_q(\mathbf{x}'; \mathbf{w}) - h_q(\mathbf{x}; \mathbf{w}) \geq 0), \tag{2}$$

where the indicator function $\mathbb{I}(\cdot)$ outputs 1 if the input is true and 0 otherwise. Note that a higher score $h_q(\mathbf{x}; \mathbf{w})$ leads to a smaller $r(\mathbf{w}; \mathbf{x}, \mathcal{S}_q)$, and a higher NDCG means a higher quality of ranking for query $q$. One can thus optimize $\mathbf{w}$ to maximize the average NDCG over all queries to obtain a good prediction score function $h_q(\mathbf{x}; \mathbf{w})$. However, doing so is challenging due to the discontinuity of $r(\mathbf{w}; \mathbf{x}, \mathcal{S}_q)$ in $\mathbf{w}$. Therefore, according to Qiu et al. (2022), one can approximate $r(\mathbf{w}; \mathbf{x}, \mathcal{S}_q)$ by a continuous and differentiable surrogate function $\bar{g}(\mathbf{w}; \mathbf{x}, \mathcal{S}_q) = \sum_{\mathbf{x}' \in \mathcal{S}_q} \ell(h_q(\mathbf{x}'; \mathbf{w}) - h_q(\mathbf{x}; \mathbf{w}))$, where $\ell(\cdot)$ is an increasing surrogate loss function of $\mathbb{I}(\cdot \geq 0)$. For example, the squared hinge loss $\ell(x) = (x + c)_+^2$. Here $c$ is a margin parameter. We define $\ell(\mathbf{w}; \mathbf{x}', \mathbf{x}, q) = \ell(h_q(\mathbf{x}'; \mathbf{w}) - h_q(\mathbf{x}; \mathbf{w}))$ and obtain a more computationally tractable **NDCG loss**:

$$L_q(\mathbf{w}) = -\frac{1}{Z_q} \sum_{\mathbf{x}_i^q \in \mathcal{S}_q} \frac{2^{y_i^q} - 1}{\log(1 + \bar{g}(\mathbf{w}; \mathbf{x}_i^q, \mathcal{S}_q))}, \tag{3}$$

and one can train $h_q(\mathbf{x}; \mathbf{w})$ by minimizing the average NDCG loss across all queries in $\mathcal{S}$.

Another loss function that measures the quality of a ranking with respect to query $q$ is the **ListNet loss** (Cao et al., 2007):

$$L_q(\mathbf{w}) = \sum_{\mathbf{x}_i^q \in \mathcal{S}_q} \left( \frac{\exp(y_i^q)}{\sum_{j=1}^{N_q} \exp(y_j^q)} \right) \cdot \log\left( \hat{g}(\mathbf{w}; \mathbf{x}, \mathcal{S}_q) \right), \tag{4}$$

where $\hat{g}(\mathbf{w}; \mathbf{x}, \mathcal{S}_q) = \sum_{\mathbf{x}' \in \mathcal{S}_q} \exp(h_q(\mathbf{x}'; \mathbf{w}) - h_q(\mathbf{x}; \mathbf{w}))$.

Given a loss function $L_q(\mathbf{w})$, e.g., (3) or (4), for each query $q \in \mathcal{Q}$, one can minimize the average loss over all queries by solving $\min_{\mathbf{w}} \frac{1}{|\mathcal{S}|} \sum_{q \in \mathcal{Q}} L_q(\mathbf{w})$ to obtain a good prediction score function $h_q(\mathbf{x}; \mathbf{w})$. However, we aim to achieve a ranking that is both high-quality and sufficiently fair. Therefore, a measure of the fairness of $h_q(\mathbf{x}; \mathbf{w})$ needs to be involved in the optimization above.

There exist multiple ways to define ranking fairness. In this paper, we focus on the equal exposure fairness of a ranking method similar to Zehlike & Castillo (2020). Formally, according to the probability distribution over $\mathcal{S}_q$ induced by scores $\{h_q(\mathbf{x}_i^q; \mathbf{w}) : \mathbf{x}_i^q \in \mathcal{S}_q\}$, the exposure of item $\mathbf{x}_i^q \in \mathcal{S}_q$ is defined as:

$$e(\mathbf{w}, \mathbf{x}_i^q, \mathcal{S}_q) := \frac{\exp(h_q(\mathbf{x}_i^q; \mathbf{w}))}{\sum_{\mathbf{x}_j^q \in \mathcal{S}_q} \exp(h_q(\mathbf{x}_j^q; \mathbf{w}))}. \tag{5}$$

The exposure in (5) can be interpreted as the reciprocal of a surrogate rank function, where higher-ranked items receive greater exposure. The reason is that we can convert it to $1/\sum_{x_j^q \in \mathcal{S}_q} \exp(h_q(x_j^q, w) - h_q(x_i^q, w))$. As a result, the denominator can be considered as a surrogate of the rank function at $x_i^q$ using the exponential surrogate function, i.e., the higher the score $h_q(x_i^q, w)$, the lower the denominator $\sum_{x_j^q \in \mathcal{S}_q} \exp(h_q(x_j^q, w) - h_q(x_i^q, w))$, matching its rank function. The exponential surrogate is widely utilized to approximate rank functions effectively (Rudin, 2009).

Given a query $q \in \mathcal{Q}$, the **equal exposure fairness** requires that the averaged exposures in both minority and majority groups be equal, namely,

$$\frac{1}{|\mathcal{S}_a^q|} \sum_{\mathbf{x}_i^q \in \mathcal{S}_a^q} e(\mathbf{w}, \mathbf{x}_i^q, \mathcal{S}_q) = \frac{1}{|\mathcal{S}_b^q|} \sum_{\mathbf{x}_i^q \in \mathcal{S}_b^q} e(\mathbf{w}, \mathbf{x}_i^q, \mathcal{S}_q). \tag{6}$$

This requirement can be satisfied by minimizing the following loss function for each query $q$, which measures the disparity in the averaged exposures between groups:

$$U_q(\mathbf{w}) := \frac{1}{2} \left[ \frac{1}{|\mathcal{S}_a^q|} \sum_{\mathbf{x}_i^q \in \mathcal{S}_a^q} e(\mathbf{w}, \mathbf{x}_i^q, \mathcal{S}_q) - \frac{1}{|\mathcal{S}_b^q|} \sum_{\mathbf{x}_i^q \in \mathcal{S}_b^q} e(\mathbf{w}, \mathbf{x}_i^q, \mathcal{S}_q) \right]^2. \tag{7}$$

The learning to rank with equal exposure fairness can be formalized as an optimization problem:

$$\min_{\mathbf{w}} \frac{1}{|\mathcal{S}|} \sum_{q \in \mathcal{Q}} L_q(\mathbf{w}) + \frac{C}{N} \sum_{q \sim \mathcal{Q}} U_q(\mathbf{w}), \tag{8}$$

where the loss $L_q(\mathbf{w})$ can be (3) or (4) or any loss function measuring the quality of ranking and loss function $U_q(\mathbf{w})$ is defined in (7), which can be viewed as a regularization term to ensuring equal exposure fairness, and the parameter $C$ balances the quality and the fairness of ranking.

## 4 Top-$K$ Ranking Fairness

The equal exposure fairness introduced in (6) is defined based on the entire output list for each query, which may not fully address unfairness in real-world applications. For example, when the ranking is used for allocating resources among disaster hotpots, the decision-makers might only prioritize the regions that are ranked at the top-$K$ position, while the ranking beyond $K$ does not matter. To extend our fairness measure to this situation, we develop a general **top-$K$ ranking fairness** metric to ensure fairness among different groups at top-$K$ positions in the output ranking. In particular, the score $h_q(\mathbf{x}; \mathbf{w})$ satisfies top-$K$ ranking fairness if

$$\frac{1}{|\mathcal{S}_a^q|} \sum_{\mathbf{x}_i^q \in S_a^q} \mathbb{I}(\mathbf{x}_i^q \in \mathcal{S}_K^q) e(\mathbf{w}, \mathbf{x}_i^q, \mathcal{S}_q) = \frac{1}{|\mathcal{S}_b^q|} \sum_{\mathbf{x}_i^q \in \mathcal{S}_b^q} \mathbb{I}(\mathbf{x}_i^q \in \mathcal{S}_K^q) e(\mathbf{w}, \mathbf{x}_i^q, \mathcal{S}_q), \tag{9}$$

where $\mathcal{S}_K^q$ denotes the set of top-$K$ items (ranked by $h_q(\mathbf{x}_i^q; \mathbf{w})$'s) in $\mathcal{S}_q$ with respect to query $q$ and $e(\mathbf{w}, \mathbf{x}_i^q, \mathcal{S}_q)$ is the exposure function in (5). As shown in (5), higher-ranked items receive higher exposure scores. When extending this to a top-$K$ ranking, each item within the top-$K$ positions is weighted according to its exposure score. The top-$K$ ranking fairness loss can be defined as the squared difference between the left and right sides of equation (9).

Ensuring the top-$K$ ranking fairness lies in the selection of items for the top-$K$ set, i.e., $\mathbf{x}_i^q \in \mathcal{S}_K^q$. Note that the top-$K$ set $\mathcal{S}_K^q$ depends on the predicted scores $h_q(\mathbf{x}_i^q; \mathbf{w})$ with model parameters $\mathbf{w}$, and a naive approach of sorting the scores will be expensive, taking $n \log n$ time complexity for a set of $n$ examples. Hence, one major innovation of this paper is to efficiently handle top-$K$ ranking fairness. Some related methods such as Wu et al. (2009); Qin et al. (2010) approximate the top-k indicator by $\psi(K - \bar{g}(w; x_i^q, S_q))$. The $\psi$ is a continuous surrogate of the indicator function. But there exists two levels of approximation errors: one in estimating the rank of an item and another in approximating $\mathbb{I}(\cdot \geq 0)$ by $\psi(\cdot)$.

To reduce approximation errors, our idea is to transform the non-differentiable top-$K$ selection operator into a differentiable one using an approach similar to Qiu et al. (2022). Specifically, let

$$\lambda_q(\mathbf{w}) = \arg\min_{\lambda} \frac{K + \varepsilon}{N_q} \lambda + \frac{1}{N_q} \sum_{\mathbf{x}_i^q \in S_q} (h_q(\mathbf{x}_i^q; \mathbf{w}) - \lambda)_+ \text{ for } q \in \mathcal{Q}, \tag{10}$$

where $\epsilon \in (0, 1)$. It can be easily proved that $\lambda_q(\mathbf{w})$ is uniquely defined and is the $K + 1$-th largest score in $\{h_q(\mathbf{x}_i^q; \mathbf{w}) | i = 1, \ldots, N_q\}$. Hence, $\mathbf{x}_i^q \in \mathcal{S}_K^q$ is selected in the top-$K$ positions if and only if $h_q(\mathbf{x}_i^q; \mathbf{w}) > \lambda(\mathbf{w})$. In other words, the indicator $\mathbb{I}(\mathbf{x}_i^q \in \mathcal{S}_K^q)$ in (9) can be replaced by $\mathbb{I}(h_q(\mathbf{x}_i^q; \mathbf{w}) - \lambda(\mathbf{w}) > 0)$.

The indicator function $\mathbb{I}(h_q(\mathbf{x}_i^q; \mathbf{w}) - \lambda(\mathbf{w}) > 0)$ is discontinuous, so we approximate it by a smooth surrogate $\psi(h_q(\mathbf{x}_i^q; \mathbf{w}) - \lambda_q(\mathbf{w}))$ (e.g., sigmoid function) within (9). The use of smooth surrogates to replace indicator functions has been studied in Bendekgey & Sudderth (2021); Yao et al. (2023). To make the loss function differentiable for optimization, we replace the absolute value in (9) with the following squared difference formulation:

$$U_q^K(\mathbf{w}, \lambda_q(\mathbf{w})) = \frac{1}{2}\left[\frac{1}{|\mathcal{S}_a^q|} \sum_{\mathbf{x}_i^q \in \mathcal{S}_a^q} \psi(h_q(\mathbf{x}_i^q; \mathbf{w}) - \lambda_q(\mathbf{w}))e(\mathbf{w}, \mathbf{x}_i^q, \mathcal{S}_q) \right.$$
$$\left. - \frac{1}{|\mathcal{S}_b^q|} \sum_{\mathbf{x}_i^q \in \mathcal{S}_b^q} \psi(h_q(\mathbf{x}_i^q; \mathbf{w}) - \lambda_q(\mathbf{w}))e(\mathbf{w}, \mathbf{x}_i^q, \mathcal{S}_q)\right]^2, \tag{11}$$

A learning to rank problem with top-$K$ ranking fairness is formulated as the following **fairness regularized bilevel optimization problem**:

$$\min_{\mathbf{w}} \frac{1}{|\mathcal{S}|} \sum_{q \in \mathcal{Q}} L_q(\mathbf{w}) + \frac{C}{N} \sum_{q \sim \mathcal{Q}} U_q^K(\mathbf{w}, \lambda_q(\mathbf{w})) \tag{12}$$
$$\text{s.t. } \lambda_q(\mathbf{w}) \text{ satisfies (10) for } q \in \mathcal{Q}.$$

However, the lower level optimization problem (10) is non-smooth and non-strongly convex, making (12) challenging to solve numerically. Hence, we approximate the lower-level problem with a smooth and strongly convex objective function. Specifically, we define:

$$\hat{\lambda}_q(\mathbf{w}) = \arg\min_{\lambda} \left\{ G_q(\mathbf{w}, \lambda) := \frac{K + \varepsilon}{N_q}\lambda + \frac{\tau_2}{2}\lambda^2 \right.$$
$$\left. + \frac{1}{N_q} \sum_{\mathbf{x}_i^q \in \mathcal{S}_q} \left[\tau_1 \ln\left(1 + \exp\left(\frac{h_q(\mathbf{x}_i^q; \mathbf{w}) - \lambda}{\tau_1}\right)\right)\right]\right\}, \tag{13}$$

where $\tau_1 > 0$ is a smoothing parameter and $\tau_2 > 0$ is a strongly convexity parameter. With this approximation, we design to solve optimization problem below:

$$\min_{\mathbf{w}} \left\{ F(\mathbf{w}) := \frac{1}{|\mathcal{S}|} \sum_{q \in \mathcal{Q}} L_q(\mathbf{w}) + \frac{C}{N} \sum_{q \sim \mathcal{Q}} U_q^K(\mathbf{w}, \hat{\lambda}_q(\mathbf{w})) \right\} \tag{14}$$
$$\text{s.t. } \hat{\lambda}_q(\mathbf{w}) \text{ satisfies (13) for } q \in \mathcal{Q}.$$

This is a challenging optimization problem for several reasons. First, an unbiased stochastic gradient of the objective function in (14) is unavailable due to the composite structure in $L_q$ and $U_q^K$. Second, when $N$ and $N_q$ are large, it is computationally expensive to update $\hat{\lambda}_q(\mathbf{w})$ for all $q$ simultaneously. To address these issues, we view (14) as an instance of the **bi-level finite-sum coupled compositional stochastic optimization** problems. It was addressed by Qi et al. (2021); Qiu et al. (2022). Then we apply a stochastic algorithm to (14). The key component of this algorithm is to construct and update the stochastic approximations of the gradients of $L_q$ and $U_q^K$ with respect to $\mathbf{w}$ using a technique called *moving average estimators*. In the next two subsections, we will provide the details on how this is done for $L_q$ and $U_q^K$, respectively.

### 4.1 Stochastic approximation for gradient of ranking loss

Let $L(\mathbf{w}) := \frac{1}{|\mathcal{S}|} \sum_{q \in \mathcal{Q}} L_q(\mathbf{w})$. When $L_q(\mathbf{w})$ is either the NDCG loss or the ListNet loss, $L(\mathbf{w})$ is a finite-sum composite function, that is,

$$L(\mathbf{w}) = \frac{1}{|\mathcal{S}|} \sum_{(q, \mathbf{x}_i^q) \in \mathcal{S}} f_{q,i}(g(\mathbf{w}; \mathbf{x}_i^q, \mathcal{S}_q)), \tag{15}$$

where $g(\mathbf{w}; \mathbf{x}_i^q, \mathcal{S}_q) = \frac{1}{N_q}\bar{g}(\mathbf{w}; \mathbf{x}_i^q, \mathcal{S}_q)$ and $f_{q,i}(g) = \frac{1}{Z_q}\frac{1-2^{y_i^q}}{\log_2(N_q g+1)}$ when $L_q$ is the NDCG loss and $g(\mathbf{w}; \mathbf{x}_i^q, \mathcal{S}_q) = \frac{1}{N_q}\hat{g}(\mathbf{w}; \mathbf{x}_i^q, \mathcal{S}_q)$ and $f_{q,i}(g) = \exp(y_i^q)/(\sum_{j=1}^{N_q}\exp(y_j^q)) \cdot \log(N_q g)$ when $L_q$ is the ListNet loss. We will only illustrate how a stochastic approximation of $\nabla L(\mathbf{w})$ can be constructed when $L_q$ is the NDCG loss because the construction for ListNet Loss is similar.

Suppose the solution at iteration $t$ is $\mathbf{w}_t$. By chain rule:

$$\nabla L(\mathbf{w}_t) = \frac{1}{|\mathcal{S}|}\sum_{(q,\mathbf{x}_i^q)\in\mathcal{S}}\nabla f_{q,i}(g(\mathbf{w}_t; \mathbf{x}_i^q, \mathcal{S}_q))\nabla g(\mathbf{w}_t; \mathbf{x}_i^q, \mathcal{S}_q). \tag{16}$$

For large-scale ranking problems, we approximate $\nabla g(\mathbf{w}_t; \mathbf{x}_i^q, \mathcal{S}_q)$ by the stochastic gradient $\nabla\hat{g}_{q,i}(\mathbf{w}_t) := \frac{1}{|\mathcal{B}_q|}\sum_{\mathbf{x}'\in\mathcal{B}_q}\nabla\ell(\mathbf{w}_t; \mathbf{x}', \mathbf{x}_i^q, q)$, where $\mathcal{B}_q$ is a subset randomly sampled from $\mathcal{S}_q$. To approximate $\nabla f_{q,i}(g(\mathbf{w}_t; \mathbf{x}_i^q, \mathcal{S}_q))$, we maintain a scalar $u_{q,i}^{(t)}$ at iteration $t$ to approximate $g(\mathbf{w}_t; \mathbf{x}_i^q, \mathcal{S}_q)$ for each query-item pair $(q, \mathbf{x}_i^q)$ and update it for iteration $t+1$ by a moving averaging scheme. That is:

$$u_{q,i}^{(t+1)} = \gamma_0\hat{g}_{q,i}(\mathbf{w}_t) + (1-\gamma_0)u_{q,i}^{(t)}, \tag{17}$$

where $\gamma_0 \in [0,1]$ is an averaging parameter. Moreover, when $|\mathcal{S}|$ is large, we also generate a subset randomly from $\mathcal{S}$, denoted by $\mathcal{B}$, and use it to approximate the average over $\mathcal{S}$ in $L(\mathbf{w})$. With these stochastic estimators, we can approximate $\nabla L(\mathbf{w}_t)$ with

$$G_1^t = \frac{1}{|\mathcal{B}|}\sum_{(q,\mathbf{x}_i^q)\in\mathcal{B}}\nabla f_{q,i}(u_{q,i}^{(t)})\nabla\hat{g}_{q,i}(\mathbf{w}_t). \tag{18}$$

## 4.2 Stochastic approximation for gradient of top-$K$ fairness regularization

Let $U(\mathbf{w}) := \frac{1}{N}\sum_{q\sim\mathcal{Q}}U_q^K(\mathbf{w}, \hat{\lambda}_q(\mathbf{w}))$. Like $L(\mathbf{w})$, $U(\mathbf{w})$ is also a finite-sum composite function but with an additional challenge that $\hat{\lambda}_q(\mathbf{w})$ is not given explicitly but through solving the lower level optimization in (13). In particular, according to (5) and (11), we have

$$U(\mathbf{w}) = \frac{1}{N}\sum_{q\in\mathcal{Q}}f_q(g_{q,a}(\mathbf{w}), g_{q,b}(\mathbf{w}), g_q(\mathbf{w})), \tag{19}$$

where

$$f_q(z_1, z_2, z_3) := \frac{1}{2}\left(\frac{z_1-z_2}{|\mathcal{S}_q|z_3}\right)^2,$$

$$g_q(\mathbf{w}) := \frac{1}{|\mathcal{S}_q|}\sum_{\mathbf{x}_j^q\in\mathcal{S}_q}\exp(h_q(\mathbf{x}_j^q; \mathbf{w})),$$

$$g_{q,a}(\mathbf{w}) := \frac{1}{|\mathcal{S}_a^q|}\sum_{\mathbf{x}_i^q\in\mathcal{S}_a^q}\psi(h_q(\mathbf{x}_i^q; \mathbf{w}) - \hat{\lambda}_q(\mathbf{w}))\exp(h_q(\mathbf{x}_i^q; \mathbf{w})),$$

$$g_{q,b}(\mathbf{w}) := \frac{1}{|\mathcal{S}_b^q|}\sum_{\mathbf{x}_i^q\in\mathcal{S}_b^q}\psi(h_q(\mathbf{x}_i^q; \mathbf{w}) - \hat{\lambda}_q(\mathbf{w}))\exp(h_q(\mathbf{x}_i^q; \mathbf{w})).$$

By chain rule, we have[1]

$$\nabla U(\mathbf{w}_t) = \frac{1}{N}\sum_{q\in\mathcal{Q}}\left[\begin{array}{l}\nabla_1 f_q(g_{q,a}(\mathbf{w}_t), g_{q,b}(\mathbf{w}_t), g_q(\mathbf{w}_t))\nabla g_{q,a}(\mathbf{w}_t)\\ +\nabla_2 f_q(g_{q,a}(\mathbf{w}_t), g_{q,b}(\mathbf{w}_t), g_q(\mathbf{w}_t))\nabla g_{q,b}(\mathbf{w}_t)\\ +\nabla_3 f_q(g_{q,a}(\mathbf{w}_t), g_{q,b}(\mathbf{w}_t), g_q(\mathbf{w}_t))\nabla g_q(\mathbf{w}_t)\end{array}\right].$$

---

[1]Here, $\nabla_k f_q$ represents the gradient of $f_q$ w.r.t. its $k$th input for $k = 1, 2, 3$.

Let $\mathcal{B}_a^q$, $\mathcal{B}_b^q$ and $\mathcal{B}_q$ be subsets randomly sampled from $\mathcal{S}_a^q$, $\mathcal{S}_b^q$ and $\mathcal{S}_q$, respectively. Suppose we have some estimators for $\hat{\lambda}_q(\mathbf{w}_t)$ and $\nabla\hat{\lambda}_q(\mathbf{w}_t)$, denoted by $\lambda_{q,t}$ and $\nabla\lambda_{q,t}$, respectively. We then approximate $\nabla g_{q,a}(\mathbf{w}_t)$, $\nabla g_{q,b}(\mathbf{w}_t)$ and $\nabla g_{q,a}(\mathbf{w}_t)$, respectively, by the stochastic gradients:

$$\nabla\hat{g}_{q,a}(\mathbf{w}_t) := \frac{1}{|\mathcal{B}_a^q|} \sum_{\mathbf{x}_i^q \in \mathcal{B}_a^q} \left(\psi'(h_q(\mathbf{x}_i^q; \mathbf{w}_t) - \lambda_{q,t}) \cdot (\nabla h_q(\mathbf{x}_i^q; \mathbf{w}) - \nabla\lambda_{q,t})\right) \exp(h_q(\mathbf{x}_i^q; \mathbf{w}))$$

$$+ \frac{1}{|\mathcal{B}_a^q|} \sum_{\mathbf{x}_i^q \in \mathcal{B}_a^q} \psi(h_q(\mathbf{x}_i^q; \mathbf{w}) - \lambda_{q,t}) \exp(h_q(\mathbf{x}_i^q; \mathbf{w})) \nabla h_q(\mathbf{x}_i^q; \mathbf{w}),$$

$$\nabla\hat{g}_{q,b}(\mathbf{w}_t) := \frac{1}{|\mathcal{B}_b^q|} \sum_{\mathbf{x}_i^q \in \mathcal{B}_b^q} \left(\psi'(h_q(\mathbf{x}_i^q; \mathbf{w}_t) - \lambda_{q,t}) \cdot (\nabla h_q(\mathbf{x}_i^q; \mathbf{w}) - \nabla\lambda_{q,t})\right) \exp(h_q(\mathbf{x}_i^q; \mathbf{w})) \tag{20}$$

$$+ \frac{1}{|\mathcal{B}_b^q|} \sum_{\mathbf{x}_i^q \in \mathcal{B}_b^q} \psi(h_q(\mathbf{x}_i^q; \mathbf{w}) - \lambda_{q,t}) \exp(h_q(\mathbf{x}_i^q; \mathbf{w})) \nabla h_q(\mathbf{x}_i^q; \mathbf{w}),$$

$$\nabla\hat{g}_q(\mathbf{w}_t) := \frac{1}{|\mathcal{B}_q|} \sum_{\mathbf{x}_j^q \in \mathcal{B}_q} \exp(h_q(\mathbf{x}_i^q; \mathbf{w})) \nabla h_q(\mathbf{x}_i^q; \mathbf{w}),$$

where $\psi'(\cdot)$ is the gradient of $\psi(\cdot)$. To approximate the gradient $\nabla_k f_q(g_{q,a}(\mathbf{w}_t), g_{q,b}(\mathbf{w}_t), g_q(\mathbf{w}_t))$, we maintain three scalars $u_{q,a}^{(t)}$, $u_{q,b}^{(t)}$ and $u_q^{(t)}$ at iteration $t$ to approximate $g_{q,a}(\mathbf{w}_t)$, $g_{q,b}(\mathbf{w}_t)$ and $g_q(\mathbf{w}_t)$, respectively, for each $q \in \mathcal{Q}$, and update them for iteration $t+1$ by a moving averaging scheme:

$$u_{q,a}^{(t+1)} = \gamma_1 \hat{g}_{q,a}(\mathbf{w}_t) + (1-\gamma_1)u_{q,a}^{(t)},$$
$$u_{q,b}^{(t+1)} = \gamma_2 \hat{g}_{q,b}(\mathbf{w}_t) + (1-\gamma_2)u_{q,b}^{(t)}, \tag{21}$$
$$u_q^{(t+1)} = \gamma_3 \hat{g}_q(\mathbf{w}_t) + (1-\gamma_3)u_q^{(t)},$$

where $\gamma_k \in [0,1]$, $k = 1, 2, 3$, are averaging parameters just like $\gamma_0$. Moreover, when $N$ is large, we also generate a subset randomly from $\mathcal{Q}$, denoted by $\mathcal{B}_Q$, and use it to approximate the average over $\mathcal{Q}$ in $U(\mathbf{w})$. Naturally, we can use the queries contained in sample $\mathcal{B} \subset \mathcal{S}$ defined in the previous subsection as $\mathcal{B}_Q$. With these stochastic estimators, we can approximate $\nabla U(\mathbf{w}_t)$ with

$$G_2^t = \frac{1}{|\mathcal{B}_Q|} \sum_{q \in \mathcal{B}_q} \left[ \begin{array}{l} \nabla_1 f_q(u_{q,a}^{(t)}, u_{q,b}^{(t)}, u_q^{(t)}) \nabla\hat{g}_{q,a}(\mathbf{w}_t) \\ +\nabla_2 f_q(u_{q,a}^{(t)}, u_{q,b}^{(t)}, u_q^{(t)}) \nabla\hat{g}_{q,b}(\mathbf{w}_t) \\ +\nabla_3 f_q(u_{q,a}^{(t)}, u_{q,b}^{(t)}, u_q^{(t)}) \nabla\hat{g}_q(\mathbf{w}_t) \end{array} \right]. \tag{22}$$

The remaining step is to approximate $\hat{\lambda}_q(\mathbf{w})$ and $\nabla\hat{\lambda}_q(\mathbf{w})$. Using the implicit function theorem as shown in Ghadimi & Wang (2018), we have $\nabla\hat{\lambda}_q(\mathbf{w}) = -\nabla_{\lambda,\mathbf{w}}^2 G_q(\hat{\lambda}_q(\mathbf{w}); \mathbf{w})(\nabla_\lambda^2 G_q(\hat{\lambda}_q(\mathbf{w}); \mathbf{w}))^{-1}$. At iteration $t$, we maintain a scalar $\lambda_{q,t}$ as an estimation of $\hat{\lambda}(\mathbf{w}_t)$, a scalar $s_{q,t}$ as an estimation of $\nabla_\lambda^2 G_q(\hat{\lambda}_q(\mathbf{w}_t); \mathbf{w}_t)$, and a scalar $v_{q,t+1}$ as an estimation of $\nabla_\lambda G_q(\lambda_{q,t}; \mathbf{w}_t)$. Let $G_q(\lambda, \mathbf{w}; \mathcal{B}_q) := \frac{K+\epsilon}{N_q}\lambda + \frac{\tau_2}{2}\lambda^2 + \frac{1}{|\mathcal{B}_q|}\sum_{\mathbf{x}_i \in \mathcal{B}_q} \tau_1 \ln(1 + \exp((h_q(\mathbf{x}_i; \mathbf{w}) - \lambda)/\tau_1))$ be an approximation of $G_q(\lambda, \mathbf{w})$ using mini-batch $\mathcal{B}_q$. We then approximate $\nabla\hat{\lambda}_q(\mathbf{w}_t)$ by $\nabla\lambda_{q,t} := -\nabla_{\lambda,\mathbf{w}}^2 G_q(\mathbf{w}_t, \lambda_{q,t}; \mathcal{B}_q)s_{q,t}^{-1}$. Then $v_{q,t+1}$ and $s_{q,t}$ are updated by a moving averaging method while $\lambda_{q,t}$ is then updated by an approximate gradient step along $v_{q,t+1}$:

$$s_{q,t+1} = (1-\gamma_4)s_{q,t} + \gamma_4\nabla_\lambda^2 G_q(\lambda_{q,t}; \mathbf{w}_t; \mathcal{B}_q),$$
$$v_{q,t+1} = (1-\gamma_4)v_{q,t} + \gamma_4\nabla_\lambda G_q(\lambda_{q,t}; \mathbf{w}_t; \mathcal{B}_q), \tag{23}$$
$$\lambda_{q,t+1} = \lambda_{q,t} - \eta_0 v_{q,t+1},$$

where $\eta_0 \geq 0$ and $\gamma_4 \in [0,1]$.

## 4.3 Algorithm and convergence result

According to the previous two subsections, we have obtained the stochastic approximations of $\nabla L(\mathbf{w}_t)$ and $\nabla U(\mathbf{w}_t)$ and thus a stochastic approximation of $\nabla F(\mathbf{w}_t)$. We then update $\mathbf{w}_t$ to $\mathbf{w}_{t+1}$ by a momentum

---

**Algorithm 1** $\underline{\text{S}}$tochastic $\underline{\text{O}}$ptmization of top-$K$ $\underline{\text{R}}$anking with $\underline{\text{E}}$xposure $\underline{\text{D}}$isparity: KSO-RED

---
1:  **for** $t = 0, \dots, T-1$ **do**
2:      Draw sample batches $\mathcal{B} \subset \mathcal{S}$ and let $\mathcal{B}_Q$ be the set of $q$'s in $\mathcal{B}$.
3:      For each $q \in \mathcal{B}_Q$, draw sample batches $\mathcal{B}_q \subset \mathcal{S}_q$, $\mathcal{B}_a^q \subset \mathcal{S}_a$, $\mathcal{B}_b^q \subset \mathcal{S}_b$
4:      **for** $(q, \mathbf{x}_i^q) \in \mathcal{B}$ **do**
5:          Compute $\hat{g}_{q,i}(\mathbf{w}_t)$ and $u_{q,i}^{(t+1)}$.
6:      **end for**
7:      **for** $q \in \mathcal{B}_Q$ **do**
8:          Compute $\nabla \hat{g}_{q,a}(\mathbf{w}_t)$, $\nabla \hat{g}_{q,b}(\mathbf{w}_t)$, $\nabla \hat{g}_q(\mathbf{w}_t)$, $u_{q,a}^{(t+1)}$, $u_{q,b}^{(t+1)}$, $u_q^{(t+1)}$, $s_{q,t+1}$, $v_{q,t+1}$ and $\lambda_{q,t+1}$.
9:      **end for**
10:     Compute $G_1^t$ and $G_2^t$ according to (18) and (22).
11:     Update $\mathbf{z}^{t+1} = (1-\gamma_5)\mathbf{z}^t + \gamma_5(G_1^t + CG_2^t)$
12:     Update $\mathbf{w}^{t+1} = \mathbf{w}^t - \eta_1 \mathbf{z}^{t+1}$
13: **end for**

---

gradient step. This procedure is presented formally in Algorithm 1, where $\mathbf{z}^{t+1}$ is the momentum gradient used to update $\mathbf{w}_t$ and $\gamma_5 \in (0,1)$ is the momentum parameter. In practice, we ignore the gradients of the top-$K$ selectors $\psi(h_q(\mathbf{x}_i^q; \mathbf{w}) - \lambda_{q,t})$ and the computation for $G_2^{t+1}$ can be simplified.

To present the convergence property of Algorithm 1, some assumptions on problem (12) are needed. We make the following assumptions on problem (12).

**Assumption 4.1.**

- $|h_q(\mathbf{x}; \mathbf{w})| \leq B_h$ for a constant $B_h$ for any $q$ and $\mathbf{x}$.

- $h_q(\mathbf{x}; \mathbf{w})$ is $C_h$-Lipschitz continuous in $\mathbf{w}$ for any $q$ and $\mathbf{x}$.

- $\nabla_{\mathbf{w}} h_q(\mathbf{x}; \mathbf{w})$ is $L_h$-Lipschitz continuous in $\mathbf{w}$ for any $q$ and $\mathbf{x}$.

- $\nabla_{\mathbf{w}}^2 h_q(\mathbf{x}; \mathbf{w})$ is $P_h$-Lipschitz continuous in $\mathbf{w}$ for any $q$ and $\mathbf{x}$.

- Stochastic gradients $\hat{g}_{q,i}(\mathbf{w})$, $\nabla \hat{g}_{q,a}(\mathbf{w})$, $\nabla \hat{g}_{q,b}(\mathbf{w})$, $\nabla \hat{g}_q(\mathbf{w})$, $\nabla_\lambda G_q(\lambda; \mathbf{w}; \mathcal{B}_q)$, $\nabla_\lambda^2 G_q(\lambda; \mathbf{w}; \mathcal{B}_q)$ and $\nabla_{\mathbf{w}\lambda}^2 G_q(\lambda; \mathbf{w}; \mathcal{B}_q)$ have bounded variance $\sigma^2$ for any $q$ and $\mathbf{x}$.

Given Assumption 4.1, similar to Qiu et al. (2022), we can prove Theorem 4.2. We have the following convergence property for Algorithm 1.

**Theorem 4.2** (Theorem 2 in Qiu et al. (2022)). *Suppose Assumption 4.1 holds and $\gamma_0, \gamma_1, \gamma_2, \gamma_3, \gamma_4, \gamma_5, \eta_0, \eta_1$ are set properly, Algorithm 1 ensures that after $T = O(\frac{1}{\epsilon^4})$ iterations we can find an $\epsilon$-stationary solution of $F(\mathbf{w})$, i.e., $\mathbb{E}[\|\nabla F(\mathbf{w}_\tau)\|^2] \leq \epsilon^2$ for a randomly selected $\tau \in \{1, \dots, T\}$.*

## 5 Experiments

### 5.1 Datasets and methods

We evaluate our algorithm in recommendation systems because several benchmark datasets are available, which contain rich item attributes such as genre and year that are useful for fairness evaluation and contain a large set of items (e.g., 20K items) for evaluating large-scale ranking systems.

**MovieLens20M** (Harper & Konstan, 2015): This dataset comprises 20 million ratings from 138,000 users across 27,000 movies. After filtering, each user has rated at least 20 movies. The dataset enriches each movie entry with metadata such as name, genre, and release year.

**Netflix Prize dataset** (Bennett et al., 2007): Originally containing 100 million ratings for 17,770 movies from 480,189 users, we use a random subset of 20 million ratings for computational feasibility, maintaining a similar structure including movie name, genre, and year.

**Derived Sensitive Group Datasets:** To evaluate fairness between protected and non-protected groups, we derive three subsets from the primary datasets:

- **MovieLens-20M-H**: horror vs. non-horror

- **MovieLens-20M-D**: documentary vs. non-documentary

- **Netflix-20M**: movies before 1990 vs. from 1990 onwards

We introduce baselines and our proposed ones, all of which are **in-processing** fair ranking frameworks.

- **K-SONG** (Qiu et al., 2022): A color-blind method achieves top-$K$ ranking accuracy without considering fairness.

- **DELTR** (Zehlike & Castillo, 2020): Optimizes ListNet ranking and disparate exposure to ensure group fairness.

- **DPR-RSP** and **DPR-REO** (Zhu et al., 2020): These methods focus on reducing ranking-based statistical parity bias and mitigating item under-recommendation bias, while maintaining recommendation performance.

- **Robust** (Memarrast et al., 2023): An adversarial learning-to-rank method that optimizes ranking utility while enforcing demographic parity fairness constraints through a minimax optimization framework.

- **NG-DE**: We integrate Qiu et al. (2022) and Zehlike & Castillo (2020) to optimize the $\underline{N}D\underline{C}\underline{G}$ ranking loss and the $\underline{D}$isparate $\underline{E}$xposure.

- **SO-RED**: Our $\underline{S}$tochastic $\underline{O}$ptimization for NDCG ranking loss and $\underline{R}$anking $\underline{E}$xposure $\underline{D}$isparity defined in 7. The fairness objective only focuses on exposure instead of top-$K$ exposure. The training objective is the same as the closest baseline DELTR. The optimization algorithm employs moving average estimators for robust gradient computation.

- **KSO-RED**: Our top-$\underline{K}$ $\underline{S}$tochastic $\underline{O}$ptimization for both top-$K$ NDCG and top-$K$ $\underline{R}$anking $\underline{E}$xposure $\underline{D}$isparity defined in 11. The code is available at: GitHub repository.

## 5.2 Evaluation metrics and experiment setup

The central aspect of our evaluation is the trade-off between accuracy and fairness. Accuracy is measured using the NDCG metric defined in 1, with higher values indicating better ranking performance. Fairness is measured by the top-$K$ exposure disparity defined in Equation 9. Specifically, for the fairness loss in the training objective, we use the Mean Squared Error (MSE) of the difference between the left and right sides of Equation 9. During evaluation, we report both MSE and Mean Absolute Error (MAE) of this difference, with smaller values indicating better fairness performance. We present MAE as the fairness loss in the main text, while the corresponding MSE results are included in the Appendix C.

For model training and evaluation, we adopt a conventional split of training, validation, and test as in Wang et al. (2020); Qiu et al. (2022). We employ the classic deep neural model NeuMF (He et al., 2017) as the prediction function $h_q(\mathbf{x}; \mathbf{w})$. For all methods, we sample the same batches of queries/users and the mixture of relevant and irrelevant items per query during each iteration to ensure a fair comparison. Details on our experimental setup are in Appendix A, including specifics on model pre-training, fine-tuning strategies, and hyperparameter selection.

For baseline models, we maintain the same hyper-parameter settings as in the original papers to ensure optimal performance and fair comparison, as detailed in Appendix B. In addition to these quantitative results, we present a visualization of how our method adjusts rankings to achieve fairer outcomes, described in Appendix F. The computing resources and computational efficiency analysis reported in Appendices D and G.

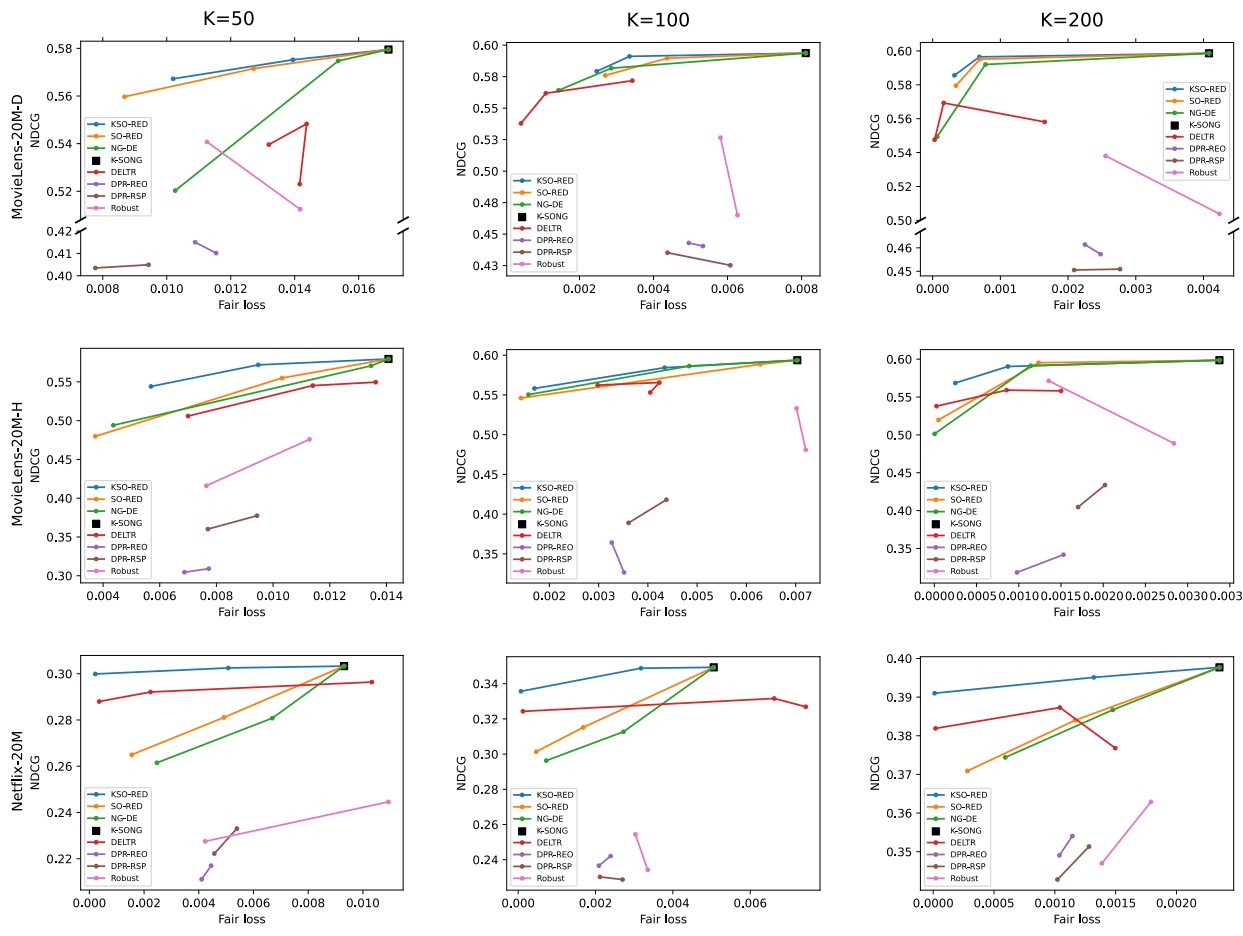

Figure 1: Comparison of accuracy and MAE fairness at top-$K$ on testing set.

## 5.3 Results

As shown in Figure 1, we evaluate a total of 7 methods on three datasets at varying top-$K$ lengths (K=50, 100, 200). In each sub-figure, the x-axis represents top-$K$ exposure disparity (the lower, the better), and the y-axis represents top-$K$ NDCG (the higher, the better). Methods are labeled with different colors. The performance of each method changes when the trade-off parameter varies. The trade-off parameter in our framework is the hyper-parameter $C$ as defined in 12 across a range from 0 to a very big number (e.g., $10^9$) to balance the quality and the fairness of ranking. A higher $C$ value places more emphasis on fairness, and $C = 0$ corresponds to a color-blind ranking algorithm with no fairness constraints, such as K-SONG.

The results indicate that our two proposed methods, in particular, KSO-RED, consistently outperform the baseline models NG-DE, DELTR, DPR-RSP, DPR-REO, and Robust across various top-$K$ results. That is, under the same fairness loss, our methods achieve the highest NDCG accuracy. Specifically, NG-DE demonstrates superior performance in terms of NDCG compared to DELTR in most scenarios, highlighting that K-SONG achieves better NDCG ranking performance than ListNet, particularly on the two MovieLens datasets. Furthermore, our SO-RED generally performs better than DELTR, which shares the same training objective, and NG-DE, validating the effectiveness of our optimization strategy. Notably, our KSO-RED, which directly focuses on top-$K$ optimization, exhibits outstanding performance at shorter top-$K$ lengths $NDCG@K(50, 100, 200)$, highlighting its proficiency in optimizing fairness within top-$K$ ranking. More details are in Appendix E.

### 5.4 Ablation Study

As shown in Figure 2, we conducted an ablation study to investigate the impact of the averaging parameter $\gamma$ in the SO-RED and KSO-RED models to investigate its impact on the trade-off between accuracy and fairness loss (MAE). For simplicity in this analysis, we set all averaging parameters $\gamma_1 = \gamma_2 = \gamma_3 = \gamma$ to the same value. We conduct experiments on the MovieLens-20M-H dataset, varying $\gamma$ from 0.2 to 1.0. The results reveal that a lower $\gamma$ value among $\{0.2, 0.6, 1.0\}$ (stronger moving average ratio) leads to a more equitable balance between accuracy and fairness, as evidenced by the top-$K$ NDCG-Fairness metrics. This trend underscores the importance of the averaging parameter in optimizing the trade-off between accuracy and fairness in recommendation systems.

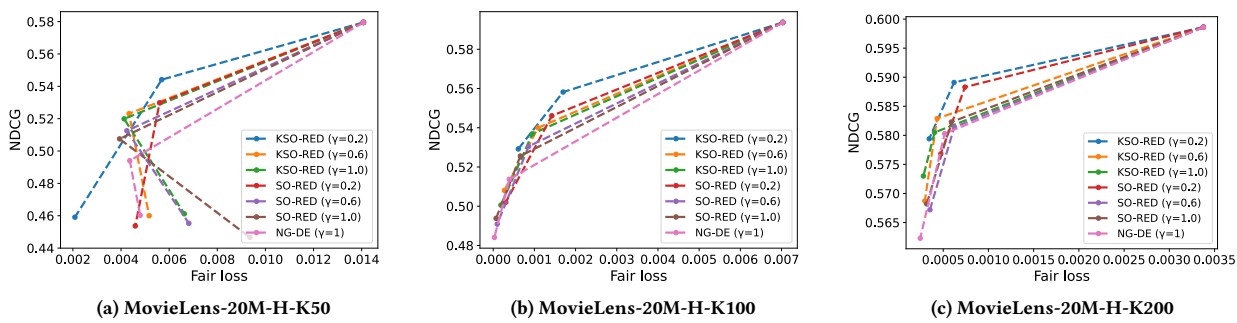

Figure 2: Comparison of different $\gamma$ values on the MovieLens-20M-H dataset.

## 6 Limitations and Future Work

First, our framework requires tuning several hyperparameters, such as the averaging parameters $(\gamma_0, \gamma_1, \gamma_2, \gamma_3, \gamma_4, \gamma_5)$, learning rate parameters $(\eta_0, \eta_1)$, and the smoothing parameters $(\tau_1, \tau_2)$. Second, our current formulation assumes binary protected attributes (e.g., minority vs. majority groups). Extending the framework to multi-group settings with more than two protected groups would require modifications to the fairness constraint formulations in (9) and corresponding algorithmic adjustments. Third, our current framework primarily focuses on optimizing exposure parity. Future work could address these limitations by incorporating adaptive hyperparameter selection methods, generalizing the approach to handle multiple protected groups, and exploring additional fairness metrics beyond exposure disparity to broaden the framework's applicability.

## 7 Conclusion

We propose a novel learning-to-rank framework that addresses the issues of inequalities in top-$K$ positions at training time. We develop an efficient stochastic optimization algorithm KSO-RED with provable convergence to optimize a top-$K$ ranking that achieves both high quality and minimized exposure disparity. Extensive experiments demonstrate that our method outperforms existing methods. Our work contributes to the development of more equitable and unbiased ranking and recommendation systems.

### Acknowledgments

This work was supported in part by the NSF under Grant No. 1943486, No. 2246757 and No. 2147253.

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

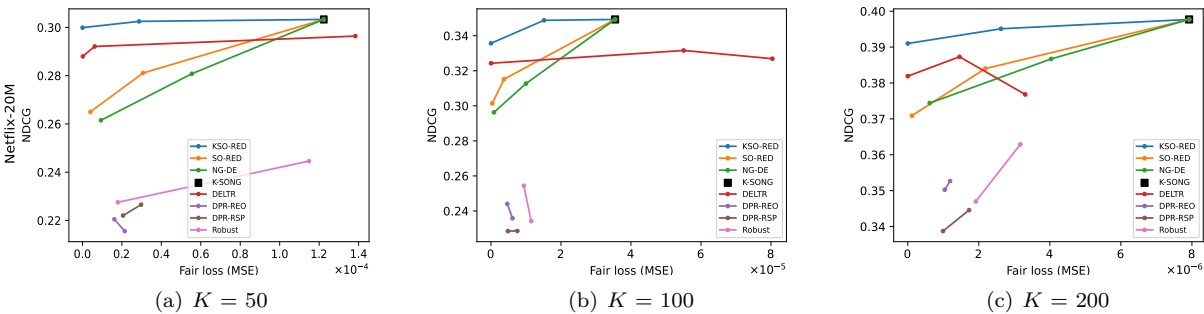

Figure 3: Comparison of accuracy and fairness loss (MSE) at top-$K$ on the Netflix-20M dataset.

# A  Experimental Details

For model training and evaluation, we adopt a conventional split of training, validation, and test as in Wang et al. (2020); Qiu et al. (2022). Specifically, we adopt the testing protocol by sampling 5 rated and 300 unrated items per user to evaluate the NDCG and fairness metrics, whereas the training employs a similar protocol as in Wang et al. (2020). For all methods, we sample the same batches of queries/users and the mixture of relevant and irrelevant items per query during each iteration to ensure a fair comparison.

The model will be pre-trained, and the resulting warm-up model will be employed by K-SONG, NG-DE, DELTR, SO-RED, and KSO-RED for a fair comparison. In our fine-tuning process, to ensure consistency and fairness in model performance comparisons, the optimal hyper-parameter settings are based on established optimal values from K-SONG. We employ the NeuMF model (He et al., 2017) as our primary predictive function due to its proven efficacy in recommendation tasks. Initially, the model undergoes a 20-epoch pre-training with a learning rate of 0.001 and a batch size of 256. Subsequent fine-tuning reinitializes the last layer, adjusting the learning rate to 0.0004 and applying a weight decay of $1 \times 10^{-7}$ over 120 epochs with a learning rate reduction by a factor of 0.25 after 60 epochs. To streamline our experiment, we leverage the tuned results from K-SONG and adopt the best value for the hyper-parameter $\gamma_0$, which is set to 0.3, serving as the base model hyper-parameter. The averaging parameters $\gamma_1, \gamma_2$, and $\gamma_3$ are tuned from the set of $\{0.2, 0.6, 1\}$.

# B  Parameters for Baseline Models

For baseline models—DELTR, DPR-RSP, DPR-REO, and Robust—we maintain the same hyper-parameter settings as used in the original papers to ensure optimal performance and a fair comparison. Specifically, in DELTR (Zehlike & Castillo, 2020) we tune the fairness hyper-parameter ($\gamma$) from 0 to 100M, covering their tuning range. For DPR-RSP and DPR-REO (Zhu et al., 2020), we select 1K and 11K for the fairness hyper-parameter, as mentioned in their respective papers. In addition, in order to adapt their code to our experimental setup, align the training/testing data with ours, and incorporate our evaluation metrics, we adjust certain data structures, such as query-item lists and batch processing of predictions to ensure their code could process large datasets effectively. All other hyper-parameters are kept the same as in the original papers. For Robust (Memarrast et al., 2023), we adapt our dataset and code to ensure a fair and consistent evaluation. Specifically, we construct query representations by generating matrices for each user and item, then concatenating user-item pair (including user, item, sensitive attributes, and the corresponding matrices) as a query for training. Given the $O(m \times n)$ time and space complexity of this method where $m$ is the number of users and $n$ is the number of items, directly applying it to large-scale datasets such as MovieLens20M and Netflix20M is impractical. To address this, we sample the training dataset, reducing the complexity to $O(m \times c)$, where $c$ is a fixed number of sampled items per user. This modification ensures scalability while maintaining the integrity of the original method. Importantly, we retain the full test dataset for evaluation,

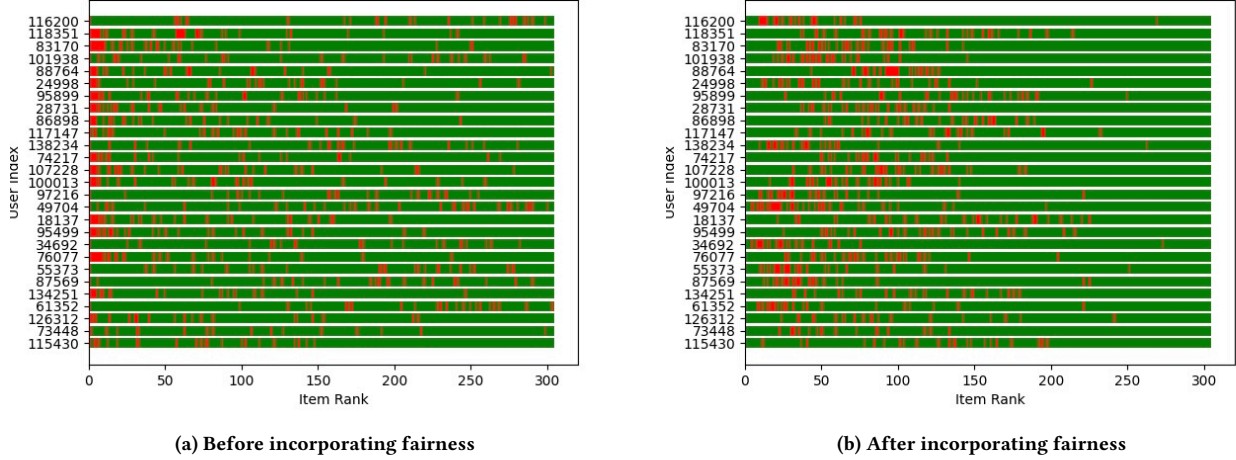

(a) Before incorporating fairness        (b) After incorporating fairness

Figure 4: Ranked item list for the most unfair 0.02% of rankings in the MovieLens-20M-H dataset.

following the original code structure to ensure a fair comparison. All other hyper-parameters are kept as specified in the original paper.

## C Fairness (MSE) vs NDCG Results

This Fig. 3 presents the corresponding fair loss (MSE) and NDCG on the Netflix-20M dataset. Similar to the MAE results shown in the main text, KSO-RED consistently outperforms the baseline models. The MSE metric provides an alternative perspective on the fairness performance by penalizing larger deviations more heavily than MAE.

## D Computing Resources for the Experiment

Our main experiments were conducted on a system equipped with the following hardware:

- 24-core Intel CPU

- 96 GB of memory

- 1 NVIDIA V100S GPU (with 32 GB memory)

- 1.5 TB SSD drive

## E Experiment Statistical Significance

Table 1 shows the NDCG and fair loss values with standard deviations for various methods across different values of $topK = 50, 100, 200$ and configurations $C = 0, 100k$. The values in parentheses represent the standard deviations for each corresponding metric, providing insight into the consistency of the performance measures.

## F Ranking Visualization

In addition to these quantitative results, we present a visualization of how our method adjusts rankings to achieve fairer outcomes, particularly on the most biased 0.02% of rankings for users in the MovieLens-20M-H dataset. Figures 4a and 4b illustrate the impact of incorporating the fairness objective during training with

| | | K | | |
|---|---|---|---|---|
| | | 50 | 100 | 200 |
| KSO-RED | C=0 | | | |
| | NDCG | 0.5769 (0.0521) | 0.5910 (0.0465) | 0.5973 (0.0430) |
| | Fair loss | 0.0139 (0.0001) | 0.0069 (0.0000) | 0.0034 (0.0000) |
| | C=100K | | | |
| | NDCG | 0.5719 (0.0541) | 0.5847 (0.0409) | 0.5901 (0.0464) |
| | Fair loss | 0.0095 (0.0000) | 0.0042 (0.0000) | 0.0012 (0.0000) |
| SO-RED | C=0 | | | |
| | NDCG | 0.5769 (0.0521) | 0.5910 (0.0465) | 0.5973 (0.0430) |
| | Fair loss | 0.0139 (0.0001) | 0.0069 (0.0000) | 0.0034 (0.0000) |
| | C=100K | | | |
| | NDCG | 0.5740 (0.0528) | 0.5886 (0.0471) | 0.5953 (0.0442) |
| | Fair loss | 0.0103 (0.0000) | 0.0063 (0.0000) | 0.0012 (0.0000) |
| K-SONG | NDCG | 0.5769 (0.0521) | 0.5910 (0.0465) | 0.5973 (0.0430) |
| | Fair loss | 0.0139 (0.0001) | 0.0069 (0.0000) | 0.0034 (0.0000) |
| NG-DE | C=0 | | | |
| | NDCG | 0.5769 (0.0521) | 0.5910 (0.0465) | 0.5973 (0.0430) |
| | Fair loss | 0.0139 (0.0001) | 0.0069 (0.0000) | 0.0034 (0.0000) |
| | C=100K | | | |
| | NDCG | 0.5707 (0.0553) | 0.5864 (0.0491) | 0.5914 (0.0465) |
| | Fair loss | 0.0135 (0.0001) | 0.0048 (0.0000) | 0.0011 (0.0000) |
| DELTR | C=0 | | | |
| | NDCG | 0.5496 (0.0551) | 0.5531 (0.0501) | 0.5581 (0.0481) |
| | Fair loss | 0.0136 (0.0001) | 0.0041 (0.0000) | 0.0015 (0.0000) |
| | C=100K | | | |
| | NDCG | 0.5452 (0.0584) | 0.5656 (0.0517) | 0.5591 (0.0484) |
| | Fair loss | 0.0114 (0.0000) | 0.0042 (0.0000) | 0.0008 (0.0000) |

Table 1: NDCG and MAE fair loss values with standard deviations for various methods.

our KSO-RED algorithm. In both figures, each row corresponds to the ranking of 305 items for a specific user, where red pixels represent items from the minority group (e.g., horror movies), and green pixels represent items from the majority group (e.g., non-horror movies).

The two figures show test set results generated by KSO-RED, trained with two different values of $C$. When $C = 0$ (Figure 4a), the rankings are color-blind, with no fairness considerations. As $C$ increases (Figure 4b), the emphasis on fairness becomes more pronounced. This visualization underscores the effectiveness of KSO-RED in adjusting rankings to ensure a more equitable distribution of exposure, particularly when higher fairness constraints are applied.

| Method | Training Time (minutes) |
|---|---|
| DELTR | 84 |
| NG-DE | 110 |
| SO-RED | 126 |
| KSO-RED and K-SONG | **127** |
| Robust | 1,522 |
| DPR-RSP/DPR-REO | >4,320 (early stop) |

Table 2: Training time comparison

# G    Computational Efficiency Analysis

We evaluate the computational efficiency of all methods on the MovieLens-20M-H dataset as an example. All experiments are conducted on the same hardware configuration (24-core Intel CPU, 96GB memory, NVIDIA V100S GPU with 32GB memory). KSO-RED, SO-RED, K-SONG, DELTR, and NG-DE are implemented under our framework with 120 training epochs, while Robust uses its original framework with 5 epochs.

Table 2 presents the training time comparison across different methods. Our proposed KSO-RED demonstrates competitive computational efficiency while achieving superior fairness-accuracy trade-offs.

