# OpenReview forum: "Learning to Rank with Top-$K$ Fairness"
_TMLR — Accepted by TMLR_

### Review · Reviewer_6jWS · 2025-07-09

**Summary Of Contributions:**

This submission addresses a critical yet underexplored issue in fairness-aware ranking: mitigating disparities in top-k exposure, where the top-ranked items significantly influence real-world decisions, such as resource allocation or emergency response. Unlike most prior work that ensures average group-level fairness across the full ranking, this paper focuses on inequities that persist at the top-k positions, where visibility and impact are greatest. The authors introduce a novel top-k exposure disparity metric, extending traditional exposure fairness to more accurately reflect real-world ranking use cases. To address the computational intractability of non-differentiable top-k selection for large item sets, the authors formulate a differentiable surrogate objective and design an efficient stochastic optimization algorithm that achieves both ranking quality and fairness. Extensive experiments demonstrate that the proposed approach outperforms existing fairness-aware ranking methods in terms of accuracy and fairness at the top-k level, providing both theoretical and empirical contributions to the study of equitable ranking systems.

**Audience:**

Yes

**Broader Impact Concerns:**

None.

**Claims And Evidence:**

Yes

**Requested Changes:**

See Weaknesses

**Strengths And Weaknesses:**

Strengths:

1. The paper is easy to follow.

2. The paper introduces a list-wise learning-to-rank framework that explicitly addresses fairness in top-k rankings during training, an area that has received limited attention in existing literature.

3. The method is rigorously validated on multiple popular benchmark datasets, demonstrating consistent improvements in both ranking accuracy and fairness compared to existing approaches.

4. The proposed method includes theoretical guarantees.

Weaknesses:

1. The optimization algorithm appears overly complex. It would be beneficial to include pseudocode in the main paper to improve clarity and reproducibility. Will authors release the code in the future?

2. Theorem 4.2 is directly taken from Theorem 2 in Qiu et al. (2022), which diminishes the novelty of the theoretical contribution.

3. The experiments are conducted solely on tabular data, leaving the effectiveness and adaptability of the proposed approach to other data modalities (e.g., image data) unclear.

4. Given the complexity of the algorithm, it would be helpful to report the training time to assess its computational efficiency.

5. A discussion on the limitations of the proposed approach would strengthen the paper.

6. While Equations (9) and (10) are non-differentiable, this does not preclude their direct incorporation into the learning objective. I recommend adding experiments to demonstrate the advantages of Equation (13) over Equations (9) and (10).

7. The experimental setup relies solely on a classic deep neural network (NeuMF). The authors are encouraged to evaluate their method using more advanced models, such as transformer-based architectures, to better assess its generalizability.

---

> ### Author Response · Authors · 2025-08-06
> **Response to Reviewer 6jWS**
>
> Thank you for your thorough review and positive feedback on our paper's clarity and contributions. We address your concerns below:
>
> **Q1:**
> The optimization algorithm appears overly complex. It would be beneficial to include pseudocode in the main paper to improve clarity and reproducibility. Will authors release the code in the future?
>
> **A1:**
> The pseudocode (Algorithm Box 1) was not displayed correctly due to formatting issues, which we have already fixed.
>
> For code release: yes, we plan to release the code upon acceptance.
>
> ---
>
> **Q2:**
> Theorem 4.2 is directly taken from Theorem 2 in Qiu et al. (2022), which diminishes the novelty of the theoretical contribution.
>
> **A2:**
> While we reference the convergence framework from Qiu et al. (2022), our theoretical contribution is substantial and non-trivial. Our approach introduces a new bilevel compositional objective that simultaneously optimizes ranking accuracy and top-K fairness. This dual-objective optimization is not addressed in Qiu et al. (2022), which focuses solely on NDCG optimization.
>
> ----
>
> **Q3:**
> The experiments are conducted solely on tabular data, leaving the effectiveness and adaptability of the proposed approach to other data modalities (e.g., image data) unclear.
>
> **A3:**
> Learning-to-rank algorithms are inherently independent of data modality. In the ranking domain, most approaches operate on ranking scores produced by any predictive function, whether from tabular, image, or text inputs. Common fairness-aware ranking approaches focus on tabular data (e.g., ratings, clicks). The predictive function can be extended to images by incorporating appropriate feature extractors. Similarly, our fairness-aware framework applies universally regardless of feature representation. However, to ensure fair comparison with other fairness papers in this domain, we focus primarily on tabular data following established practices in the field.
>
> ---
>
> **Q4:**
> Given the complexity of the algorithm, it would be helpful to report the training time to assess its computational efficiency.
>
> **A4:**
> We have added the time report in appendix F Computational Efficiency Analysis.
>
> ---
>
> **Q5:**
> A discussion on the limitations of the proposed approach would strengthen the paper.
>
> **A5:**
> We have already added a Limitations and Future Work section covering key constraints of our approach, including:
> - Hyperparameter tuning: Performance depends on tuning of averaging parameters (γ₀, γ₁, γ₂, γ₃) and smoothing parameter (τ₁)
> - Binary group assumption: Current formulation assumes binary protected attributes, extension to multi-group settings needs changes in some formulas.
>
> ---
>
> **Q6:**
> While Equations (9) and (10) are non-differentiable, this does not preclude their direct incorporation into the learning objective. I recommend adding experiments to demonstrate the advantages of Equation (13) over Equations (9) and (10).
>
> **A6:**
> Direct incorporation of Equations (9) and (10) is not applicable. Equation (9) requires combinatorial optimization over ranking permutations at each step, which doesn't scale to our large datasets (20M+ samples). Equation (10) lacks the theoretical support to solve. In contrast, for (13), we can appeal to the implicit function theorem to estimate gradients.
>
> ---
>
> **Q7:**
> The experimental setup relies solely on a classic deep neural network (NeuMF). The authors are encouraged to evaluate their method using more advanced models, such as transformer-based architectures, to better assess its generalizability.
>
> **A7:**
> Our optimization framework can be used with any predictive scoring function, such as NeuMF model or a transformer model. In this paper, we focus primarily on fairness-aware optimization rather than a specific predictive function. To ensure fair comparison, we adopt a consistent score predictive function (NeuMF) across most baselines, including K-SONG, NG-DE, DELTR.

---

> > ### Comment · Reviewer_6jWS · 2025-08-12
> > **Response**
> >
> > Thank you for your response.
> >
> > I am not satisfied with the responses to my Q3, Q6, and Q7.
> >
> > For Q3, I still expect to see an evaluation of the proposed method’s adaptability to other data modalities, particularly image data. Without such experiments, the impact of your approach will appear limited.
> >
> > For Q6, I do not agree with the statement that it “does not scale to your large datasets.” Setting aside the theoretical justification, you could directly use Eq. (9) or Eq. (10) in the experiments with SGD. I do not see any significant challenges in obtaining performance results.
> >
> > For Q7, NeuMF is too specific. To better demonstrate the general effectiveness of your method, I still recommend testing it with different model architectures.

---

> > > ### Author Response · Authors · 2025-08-13
> > > **Response to Reviewer 6jWS**
> > >
> > > We are confused by the reviewer's suggestion to “directly use Eq. (9) or Eq. (10) in the experiments with SGD.” Below, we interpret his/her comment in the most reasonable way we can and explain why such an approach is infeasible.
> > >
> > > Regarding Eq. (9):
> > > We assume the reviewer means replacing $U^K_q$​ in (8) with the violation measure in (9) and then solving (8) via SGD. However, Eq. (9) is not only non-differentiable — it is discontinuous due to the 0–1 indicator function. We are not aware of any work that applies SGD directly to such a discontinuous objective, as the gradient is not defined at points of discontinuity. In practice, SGD requires at least subgradient information, which is unavailable here.
> > >
> > > Regarding Eq. (10):
> > > We assume the reviewer means solving (12) directly, where the lower-level subproblem is (10). This is a bilevel optimization problem, for which standard SGD is not applicable. A common approach is to approximate the gradient via the implicit function theorem, which requires inverting the Hessian of (10) with respect to $\lambda$. Unfortunately, (10) is piecewise linear in $\lambda$, so its Hessian is zero almost everywhere and therefore not invertible. This not only has no the theoretical justification but also leads to numerical error by inverting a zero matrix.

---

### Review · Reviewer_2bKo · 2025-07-20

**Summary Of Contributions:**

This paper proposes a list-wise learning-to-rank framework to improve fairness at the top-$K$ positions in ranking tasks, motivated by the real-world importance of resource allocation and exposure disparities for protected groups. The main contributions are (i) developing a new top-$K$ exposure disparity measure extending classical fairness notions, (ii) integrating this measure as a regularizer within a differentiable and scalable optimization framework using stochastic approximation, and (iii) demonstrating via experiments on standard recommendation datasets that the proposed approach can significantly reduce unfairness at the top-$K$ positions while retaining high ranking accuracy.

**Audience:**

Yes

**Broader Impact Concerns:**

NA.

**Claims And Evidence:**

Yes

**Requested Changes:**

- You may consider including additional baselines (especially modern transformer-based ranking architectures and recent fairness-aware methods) and evaluating at least one other domain (e.g., search, hiring, credit lending) to bolster the claimed generality of the approach.
- The use of a sigmoid surrogate to replace the indicator function in the fairness literature has been investigated in [1-2]. You may consider citing these papers in appropriate positions, such as on page 5.
- The format of the pseudocode is incorrect.
- You don't need to use two lines here in Eq. (9).

[1] Scalable and Stable Surrogates for Flexible Classifiers with Fairness Constraints. NeurIPS 2021.
[2] Understanding Fairness Surrogate Functions in Algorithmic Fairness. TMLR 2024.

**Strengths And Weaknesses:**

Strengths
- The claims made in the paper are generally well supported by both theoretical derivations and experimental evidence. The central claim—that the proposed KSO-RED method effectively reduces top-$K$ exposure disparity while maintaining ranking accuracy—is substantiated by empirical studies across several datasets (MovieLens-20M-H, MovieLens-20M-D, Netflix-20M)
- The methods are clearly described, with the progression from classical NDCG and exposure fairness to the top-$K$ extension articulated step-by-step.

Weaknesses
- Proof details of Theorem 4.2 are omitted and mostly referenced to prior work (Qiu et al., 2022). The Algorithm and convergence result mainly follow a standard approach, and the theory has been well-developed in previous work.
- For broader impact, evaluation on, for instance, real-world demographic attributes or additional public datasets (e.g., web search, recruitment) may be considered.
- More ablation study is needed for key hyperparameters.

I don't have many major comments on this paper. It appears to follow a standard methodology, and the core idea seems straightforward. For example, the optimization objective, Top-K selection operator transformation, gradient derivations, and theoretical results align closely with established approaches. While these results are correct, they don't strike me as particularly surprising. The primary contribution appears to be the adaptation and connection of existing ideas from previous work to address the Top-K ranking fairness problem. I'm somewhat uncertain about the overall significance of this specific approach within the field.

---

> ### Author Response · Authors · 2025-08-06
> **Response to Reviewer 2bKo**
>
> Thank you for your constructive feedback. We appreciate your acknowledgement that our claims are well-supported and methods are clearly described. We address your requested weaknesses and changes below:
>
> **Q1:**
> Proof details of Theorem 4.2 are omitted and mostly referenced to prior work (Qiu et al., 2022). The Algorithm and convergence result mainly follow a standard approach, and the theory has been well-developed in previous work.
>
> **A1:**
> The proof of algorithm convergence is similar to Qiu et al. (2022). Section 4.2 provides details on approximating the fairness objective G2. Since the assumptions about the function properties of G2 are similar to those for G1 (as discussed in Section 4.1 and used in Qiu et al., 2022), the proof in Section 4.3 follows a similar structure. However, our theoretical contribution remains substantial and non-trivial (see also response A4 for further discussion).
>
> ----
>
> **Q2:**
> - For broader impact, evaluation on, for instance, real-world demographic attributes or additional public datasets (e.g., web search, recruitment) may be considered.
> - You may consider including additional baselines (especially modern transformer-based ranking architectures and recent fairness-aware methods) and evaluating at least one other domain (e.g., search, hiring, credit lending) to bolster the claimed generality of the approach.
>
> **A2:**
> Our model focuses primarily on optimizing fairness-aware objectives. We have included all applicable recent work that focuses on fairness, encompassing traditional learning-to-rank approaches (DELTR), debiasing methods (DPR-RSP, DPR-REO), adversarial approaches (Robust), top-K optimization (K-SONG), and combined approaches (NG-DE).
>
> While transformer-based scoring functions are a recent alternative, we use the NeuMF predictive function to ensure a fair comparison with existing baselines.
>
> Regarding additional datasets, we have searched extensively for datasets with real demographic attributes, such as those in recruitment domains. However, such datasets are scarce due to privacy concerns. Some well-known recommendation datasets (e.g., RecSys Challenge 2017 XING) have expired. We will seek more diverse datasets.
>
> ----
>
> **Q3:**
> More ablation study is needed for key hyperparameters.
>
> **A3:**
> We have already included an ablation study in Appendix E previously, demonstrating the impact of tuning the most fairness-relevant hyperparameter γ. In the revision, we move it to 5.4 in the main text. The other hyperparameters are detailed in Appendix A. We can incorporate additional ablation study experiments and results if needed.
>
> ----
>
> **Q4:**
> “The primary contribution appears to be the adaptation and connection of existing ideas from previous work to address the Top-K ranking fairness problem”
>
> **A4:**
> While the convergence proof builds on prior work, our key contribution lies in formulating and solving the novel problem of in-processing top-K fairness optimization. Our approach introduces a new bilevel compositional objective that simultaneously optimizes ranking accuracy and top-K fairness. This dual-objective optimization is not addressed in Qiu et al. (2022), which focuses solely on NDCG optimization.
>
> ----
>
> **Q5:**
> The use of a sigmoid surrogate to replace the indicator function in the fairness literature has been investigated in [1-2]. You may consider citing these papers in appropriate positions, such as on page 5.
>
> **A5:**
> Thank you for pointing out these relevant works.
>
> We have already added these citations on page 5 where we discuss the smooth surrogate function ψ(·).
>
> ----
>
> **Q6:**
> The format of the pseudocode is incorrect.
>
> **A6:**
> The algorithm was not displayed correctly due to formatting issues, which we have already fixed in the revision.
>
> ---
>
> **Q7:**
> You don't need to use two lines here in Eq. (9).
>
> **A7:**
> We have already reformatted Equation (9).

---

> > ### Comment · Reviewer_2bKo · 2025-08-08
> >
> > Thank you for your detailed and thoughtful responses. I have no further questions at this stage.
> >
> > To further enhance the manuscript, I suggest you more explicitly highlight the key contributions and novelty of your work. Specifically, consider strengthening the discussion around how your proposed dual-objective optimization addresses the existing limitations of Top-K fairness problems, or underscore the importance and lack of prior solutions to the problem you tackle. This will help readers better appreciate the significance of your work.

---

### Review · Reviewer_cXtn · 2025-07-28

**Summary Of Contributions:**

The paper introduces an in-processing learning-to-rank framework that targets fairness within the top K positions by defining a top K exposure disparity metric and jointly optimizing for both relevance and fairness. To make the hard top K selection differentiable, the authors replace the non-differentiable indicator with a smooth surrogate and formulate a bilevel objective, resulting in an efficient and trainable approximation. They develop a stochastic compositional optimizer with momentum, KSO-RED, and provide a convergence guarantee to an $\epsilon$-stationary point under standard smoothness and bounded-variance assumptions, building on techniques from K-SONG. The approach is scalable, does not require group labels at test time, and the optimization framework can be extended to other list-wise fairness notions.

**Audience:**

Yes

**Claims And Evidence:**

Yes

**Requested Changes:**

1. Could the authors elaborate on why KSO‑RED and SO‑RED perform so similarly on MovieLens‑20M‑H? Clarifying when top‑$K$ fairness meaningfully differs from standard (full‑list) exposure fairness, and thus when the proposed method should be preferred, would help readers decide which approach to use.

2. Equation (9) appears to state a fairness constraint, not a loss. Should the fairness loss be defined as the absolute difference between the group exposures instead? Please clarify the exact loss used in Figure 1.

3. I believe a $\lambda$ term is missing in the first term of Equation (10); without it, the expression seems inconsistent with the surrounding derivation. Please verify and correct.

**Strengths And Weaknesses:**

Strengths:

1.	The paper addresses an underexplored problem, ensuring fairness specifically in the top K-ranked items by introducing a principled top K exposure disparity metric.

2.	The proposed bilevel optimization with a smooth surrogate and the KSO RED optimizer is both theoretically grounded (with convergence guarantees) and practically scalable to large real-world datasets.

Weaknesses:

1.	Insufficient Practical Motivation: While the problem is well formulated, the paper lacks a compelling justification for why top K fairness is particularly critical in practice. A stronger empirical or application-driven motivation would strengthen the case.

---

> ### Author Response · Authors · 2025-08-06
> **Response to Reviewer cXtn**
>
> Thank you for your thoughtful review and constructive feedback. We appreciate your recognition of our novel problem formulation and theoretical contributions. We address your requested changes below:
>
> **Q1:**
> Insufficient Practical Motivation: While the problem is well formulated, the paper lacks a compelling justification for why top K fairness is particularly critical in practice. A stronger empirical or application-driven motivation would strengthen the case.
>
> **A1:**
> Thank you for the thoughtful feedback. Top-K fairness is particularly critical in many real-world applications where only the highest-ranked items receive attention or resources. For example, in hiring, only the top-K candidates may be interviewed or considered further, making it essential to ensure fair representation at the top of the ranking [1]. Similarly, in disaster response or predictive policing [2], top-K hotspot rankings determine where limited resources (e.g., emergency personnel or patrol units) are deployed. In such settings, fairness in the top-K results can have a direct and significant real-world impact. We can highlight application-driven examples in the paper introduction section.
>
> [1] Kweon, Wonbin, et al. "Top-personalized-K recommendation." Proceedings of the ACM Web Conference 2024. 2024.
>
> [2] Mohler, George, et al. "A penalized likelihood method for balancing accuracy and fairness in predictive policing." 2018 IEEE international conference on systems, man, and cybernetics (SMC). IEEE, 2018.
>
> ---
>
> **Q2:**
> Could the authors elaborate on why KSO‑RED and SO‑RED perform so similarly on MovieLens‑20M‑H? Clarifying when top‑ fairness meaningfully differs from standard (full‑list) exposure fairness, and thus when the proposed method should be preferred, would help readers decide which approach to use.
>
> **A2:**
> Thank you for this insightful question. The performance is similar on MovieLens-20M-H  because of relatively low fairness constraints (small hyperparameter $C$ values). Both methods achieve fairness with comparable accuracy drops. However, the key difference emerges in Netflix-20M, when a much larger $C$ value is required to achieve near-perfect fairness (fair loss near 0). In general, KSO-RED's direct top-K optimization provides more precise control over the top positions. When fairness requirements become strict, this advantage becomes clearer.
>
> ---
>
> **Q3:**
> Equation (9) appears to state a fairness constraint, not a loss. Should the fairness loss be defined as the absolute difference between the group exposures instead? Please clarify the exact loss used in Figure 1.
>
> **A3:**
> Equation (9) clarification:
> You are correct. Equation (9) defines a fairness constraint (equality condition), not a loss function. The actual loss function is the absolute difference between the left and right sides of Equation (9). When this loss equals zero, it indicates perfect fairness. In Figure 1, the fairness loss is calculated based on this absolute difference from Equation (9).
>
> We have already revised this in the paper for clarity.
>
> ---
>
> **Q4:**
> I believe a  term is missing in the first term of Equation (10); without it, the expression seems inconsistent with the surrounding derivation. Please verify and correct.
>
> **A4:**
> Thank you for identifying this typographical error. The correct formulation should be:
> $\lambda_q(w) = \arg\min_\lambda \frac{K+\varepsilon}{N_q}\lambda + \frac{1}{N_q}\sum_{x_i^q \in S_q}(h_q(x^q_i; w) - \lambda)_+\text{ for }q \in Q$
>
> The λ coefficient was missing (we missed it in the paper writing). The other equations around it are correct. We have already corrected this typo in the revision.

---

### Decision · Action_Editor_WHps · 2025-09-09

**Recommendation:** Accept with minor revision

**Additional Comments:**

This paper studies a new setting where the fairness is considered among the top k ranked candidates. The key contribution lies in formulating and solving the novel problem of in-processing top-K fairness optimization. The method introduces a new bilevel compositional objective that simultaneously optimizes ranking accuracy and top-K fairness.

The reviewers acknowledged the strength of the paper, including a new problem setting, theoretically-grounded bilevel optimization formulation, clearly described method, experiments on multiple benchmarks. There are also some concerns like the significance of the problem is not well motivated by real problem and the contribution and novelty should be highlighted. According to TMLR policy, I think the paper is technically sound and has potential audience. I thus recommend acceptance.

Suggested minor revision must include the following point:
after addressing the definition issue of the loss function in Equation (9), the authors write in the revised version: “The top-K ranking fairness loss can be defined as the squared difference between the left and right sides of Equation (9).” However, in Section 5.2 they state: “Fairness is measured by the top-K exposure disparity defined in (9), i.e., the absolute difference between the left and right sides of Equation (9).” Such inconsistencies are frustrating and raise concerns about the paper’s rigor.

**Audience:**

Yes

**Audience Explanation:**

It would be interesting to researchers and practitioners who care about fairness in their ranking systems.

**Claims And Evidence:**

Yes

**Claims Explanation:**

The effectiveness of the proposed method is verified by experiments.